# The Influence of Analgesic Wound Infiltration on Postoperative Pain and Inflammatory Cytokines in Open Colorectal Surgery: A Randomized Comparative Pilot Study

**DOI:** 10.3390/medicina60081244

**Published:** 2024-07-31

**Authors:** Raluca Cristina Ailioaie, Elena Stefanescu, Crina Leahu, Alexandra Boldis, Razvan Scurtu

**Affiliations:** 1Department of Surgery, Iuliu Hatieganu University of Medicine and Pharmacy, 400012 Cluj-Napoca, Romania; raluca.apostu@umfcluj.ro; 2Anesthesia and Intensive Care Department, Emergency Clinical County Hospital, 400006 Cluj-Napoca, Romania; stefanescuelena2004@yahoo.com (E.S.); eahu.crina.elena@elearn.umfcluj.ro (C.L.); boldis.alexandra.maria@elearn.umfcluj.ro (A.B.)

**Keywords:** wound infiltration, local anesthetic, interleukins, Il-6, Il-10, colorectal surgery

## Abstract

*Background and Objectives:* Surgical wound analgesia has been analyzed in many studies, but few have focused on its relationship with inflammatory markers. As such, we aimed to determine the influence of analgesic surgical wound infiltration in open colorectal surgery on the seric levels of pro- and anti-inflammatory markers and the associated efficacy in postoperative pain control. *Materials and Methods:* Forty patients who underwent open colorectal surgery were prospectively randomized: group 0, epidural analgesia; group 1, intravenous analgesia (control), group 2, preincision and prelaparoraphy infiltration; and, group 3, prelaparoraphy infiltration. Wound infiltration was performed with ropivacaine. We analyzed the levels of IL-6 and IL-10 cytokines before and 6 h after surgery and their correlation with pain scores. *Results:* The postoperative Il-6 levels were significantly lower in group 0 than in the control (*p* = 0.041). The postoperative Il-10 levels were significantly higher in group 3 (*p* = 0.029) than in the control. Six hours after the operation, the pain scores were significantly lower in all groups than in the control (*p* = 0.005, *p* = 0.022, and *p* = 0.017 for groups 0, 2, and 3, respectively). Pain scores were significantly correlated with Il-10 levels in group 2 (*p* = 0.047); in group 3, IL-10 levels directly correlated with those of Il-6 (*p* = 0.026). *Conclusions:* The analgetic effect of preincisional and prelaparoraphy analgetic infiltration was efficient. The analgetic infiltration of the surgical wound prior to closure stimulates both the inflammatory activator and regulator interleukins.

## 1. Introduction

Open colorectal surgery is associated with severe and prolonged postoperative pain, which is augmented by the patient’s active mobilization [1]. Inadequate pain control increases postoperative morbidity; therefore, adequate treatment is essential for reducing the systemic response and, thus, the rates of complications and the length of hospital stay, as well as increasing patient satisfaction [2,3]. Effective postoperative pain management is essential for reducing the inflammatory response and the onset of organ dysfunction. A multimodal strategy, of which the main component is locoregional analgesia, represents the current standard approach to pain management [2].

Surgical trauma substantially changes the inflammatory system of the host organism [2]. The complement system is activated, and pro- and anti-inflammatory cytokines are released during and after major surgical interventions [4]. Cytokines are intracellular proteins with a low molecular weight and high potency, which play important roles through specific receptors in modulating the immune response, hematopoiesis, and the inflammatory changes that occur during and after surgery [5]. The body’s response to surgical injury leads to the migration of inflammatory cells, with the release of cytokines, especially IL6, generating local inflammatory reactions. Cytokines enhance the generalized inflammatory reaction when they reach the systemic circulation, activating C-reactive protein and other proinflammatory markers (TNF alpha, IL-6, IL1beta, IL 1ra, and IL-8). Later, a compensatory anti-inflammatory response occurs, which is noted by an increase in the levels of anti-inflammatory cytokines such as IL-10 [6].

The surgical incision is one of the main sources of postoperative pain [7]. The infiltration of the surgical incision with a local anesthetic is associated with reduced nociception and pain transmission, playing a major role in the patient’s fast postoperative recovery. The infiltration of the surgical wound with local anesthetic (intermittent or continuous) at the end of the intervention has been described in numerous studies and has frequently been compared with epidural analgesia or with the postoperative use of opioids. However, the results are inconsistent [3,8]. The heterogeneity of these results can be attributed to the samples (patients undergoing different operations and/or different surgical techniques), the administration of local anesthetic, local anesthetic dosage, as well as the overall postoperative analgesic management [9]. Here, pre-emptive analgesia, which is defined as the antinociceptive intervention administered before the appearance of pain stimulus, plays an important role [10].

Epidural analgesia is part of the multimodal management of postoperative pain and is included in enhanced surgery recovery protocols. The benefits of this method have been proven in many studies [11,12,13]. The administration of an epidural in open colorectal surgery is considered standard but is associated with complications related to the technique, as well as numerous side effects [11]. Additionally, epidural analgesia does not decrease the rate of postoperative cardiopulmonary complications or the length of hospital stay [12,14]. In this context, alternative methods of pain management should be available.

Given the inconsistent results presented in the literature, we conducted a pilot study that compared multimodal analgesia strategies and focused on pain management in correlation with the inflammatory biologic response.

## 2. Materials and Methods

### 2.1. Study Protocol

This experimental protocol was designed in accordance with Consolidated Standards of Reporting Trials (CONSORT) and approved by the Ethics Committee of Iuliu Hatieganu University of Medicine and Pharmacy No. 61/1 March 2022 and Ethics Committee of Emergency Clinical County Hospital Cluj-Napoca No. 1603/13 January 2021. Written consent was obtained from all subjects.

### 2.2. Inclusion and Exclusion Criteria

This study was conducted in the First Surgical Clinic, Emergency County Hospital Cluj-Napoca, between February 2021 and November 2023. We performed a prospective randomized pilot study that included 40 patients. The inclusion criteria were patients aged between 18 and 80 years old, diagnosed with colorectal cancer (CRC), and scheduled for elective open surgery with our service, with an ASA risk score of I-III. The exclusion criteria included a history of allergic reaction to local anesthetics or analgesics (paracetamol, tramadol, or morphine); an ASA risk > III for patients with organ failure for which the administered dose of local anesthetic should have been reduced; moderate or severe hepatic or renal failure; patients with inflammatory bowel disease; psychiatric disorders, or addiction to opioids or other analgesics; or refusal of the patient. Patients with surgical complications in the first 24 h after the surgery were also excluded from this study.

### 2.3. Randomization and Wound Infiltration Technique

All patients were evaluated regarding the indication for epidural analgesia (group 0) as a standard analgesic technique. The subjects not included in this group, because of refusal, inflammation/infection at the puncture site, severe mitral or aortic stenosis, intracranial hypertension, history of spinal surgery, neurological lesions, or failure of the procedure, were randomized into three other groups: control group (group 1), pre- and postoperative wound infiltration (subcutaneously at the incision site and in the preperitoneal space) (group 2), and postoperative wound infiltration (preperitoneal space) (group 3). Patients were randomized according to the block randomization method.

The control group (group 1) received multimodal analgesic treatment according to hospital protocol, without wound infiltration. The multimodal analgesia was represented by Paracetamol, Dexamethasone, Ketamine, Nefopam, and Metamizole.

We used 0.375% ropivacaine as a local anesthetic adjusted to patient weight, with a maximum dose of 2 mg/kg.

The epidural group (group 0) received continuous infusion with 0.2% ropivacaine on the day of the surgery, followed by 0.1% starting the first day after surgery and continuing for 48 h.

In group 2, wound infiltration was performed before and after the surgery. A total of 20 mL of 0.375% ropivacaine was subcutaneously infiltrated at the incision site, and another 20 mL was infiltrated in the preperitoneal space before the closure of the abdominal wall.

In group 3, wound infiltration was performed only in the preperitoneal space, along the incision, before the closure of the abdominal wall, with 20 mL of 0.375% ropivacaine.

### 2.4. Groups Analysis

All patients with proven CRC underwent open surgery through midline laparotomy. For colon cancer, a right hemicolectomy, left hemicolectomy, or sigmoid resection was performed, whereas for rectal cancer, anterior resection or Hartmann resection was performed, depending on tumor and patient characteristics. A loop ileostomy was associated as a part of the surgical procedure in the low anterior resection of the rectum.

### 2.5. Anesthetic and Analgesic Technique

All patients received a balanced general anesthesia, with standard monitorization (arterial blood pressure, EKG, peripheral O_2_ saturation, capnography, train-of-four (TOF), and temperature). One hour before induction, all patients received 1 g of paracetamol i.v. and prophylactic antibiotics. The induction and maintenance of the anesthesia were coordinated by the anesthesiologist with intraoperative multimodal analgesia according to hospital protocol. Postoperative analgesia was also implemented according to hospital protocol.

The efficiency of the analgesia was evaluated after surgery using a visual analog scale (VAS) for pain, where 0 indicates no pain; 1–3 indicate easy pain; 4–7 represent moderate pain; and 8–10 indicate severe pain, at different time intervals: 6, 12, 18, and 24 h after surgery.

For pain scores > 4, rescue therapy with opioids was used (tramadol or morphine i.v.).

### 2.6. Inflammatory Markers

Venous blood samples were obtained just before anesthetic induction or before wound infiltration (T0) and 6 h after the surgical procedure (T6) to determine the levels of the inflammatory markers. Blood samples were used to determine the levels of C reactive protein (CRP) and leucocytes, whereas for cytokine identification, blood was centrifuged within one hour at 1000 rpm for 5 min, and plasma was preserved at −20 °C until analysis. On the same day, the samples were sent to a specialized laboratory for the quantitative determination of interleukin 6 (Il-6) and interleukin 10 (Il-10) levels.

We analyzed the levels of IL-6 and IL-10 cytokines at T0 and T6 and the dynamics of CRP and leukocytes. Pain scores were recorded 6, 12, 18, and 24 h after surgery. Additionally, the length of hospital stay (LHS) and duration of admission in the intensive care unit (ICU) were recorded, as well as postoperative morbidity. A comparative analysis was performed between the groups.

### 2.7. Statistical Analysis

Every marker, including length of hospital stay, was analyzed using descriptive statistics. Values were generally compared between the groups using a unifactorial ANOVA test, whereas multiple comparisons were performed with the post hoc Bonferroni test. The pain scores were analyzed using the Kruskal–Wallis and post hoc Duncan’s tests, which were followed by the median test. Comparisons within groups were performed using the *t* test for pairs. To determine correlations, the Pearson correlation coefficient was used. Statistical significance was considered at a *p* value < 0.05. Data were analyzed using IBM SPSS Statistics version 22.0.

## 3. Results

The main characteristics of the groups are illustrated in Table 1.

The mean levels of the evaluated biomarkers, before and at 6 h after surgery, are reported in Table 2a. A significant increase in postoperative Il-6 levels was noticed in all groups (Table 2b). In groups 0, 2, and 3, the differences between T0 and T6, although significant, were smaller than those of the control group. A significant difference was also observed in the Il-10 levels in group 3 between T0 and T6 (Table 2a).

In a comparative analysis of the cytokine Il-6 levels among all the groups, a statistically significant difference was found in the postoperative Il-6 levels (*p* = 0.018, Table 2b). In the epidural group (128.07 ± 138.29), the postoperative Il-6 levels were significantly lower than in the control (435.54 ± 340.36; *p* = 0.041; Figure 1). In group 2, the mean postoperative Il-6 levels (371.91 ± 317.24) were lower than that in the control but higher than that in the epidural group; in group 3, the mean postoperative Il-6 levels (147.06 ± 140.91) were lower than that in the control and closer to that in the epidural group, although this difference was not statistically significant.

No statistically significant differences were found between the groups regarding pre-operative Il-6 or Il-10 (*p* = 0.18, *p* = 0.92), postoperative Il-10 (*p* = 0.64), pre- and postoperative CRP (*p* = 0.91, *p* = 0.53), or leukocyte (*p* = 0.53, *p* = 0.3) levels.

A statistically significant difference was noticed in the pain scores at T6 between the control group and the other groups: group 0 (*p* = 0.005), group 2 (*p* = 0.022), and group 3 (*p* = 0.017) (Figure 2).

A significant indirect correlation was found between the pain scores and Il-10 levels at T6 (Figure 3) in group 2 (R = −44.8%, *p* = 0.047). A direct correlation was also found between postoperative cytokine Il-6 and Il-10 levels (R = 66.4%, *p* = 0.026) in group 3 (Figure 4).

The mean hospitalization duration for each group is listed in Table 3. No differences were identified regarding the total LHS or stay in the ICU (*p* = 0.33, *p* = 0.34) between the groups.

We found no statistically significant differences between the groups in opioid use, as shown in Table 4.

During admission, complications were registered in group 0 or 3. In the epidural group, four patients developed pneumonia, ileus, pelvic abscess, or chyloperitoneum. In the preperitoneal infiltration group, one patient developed subcutaneous hematoma, unrelated to the infiltration site, which was preperitoneal.

## 4. Discussion

We performed this study to evaluate the effect of different wound infiltration techniques on postoperative pain. We also aimed to determine whether better postoperative pain control correlated with lower proinflammatory and higher anti-inflammatory cytokine levels.

We chose interleukins 6 and 10 as proinflammatory and anti-inflammatory markers, respectively. They are involved in many different pathways in the organism, and their levels can be influenced by many factors. These interleukins strongly influence the immunity response in CRC. Il-6 regulates the cellular immune response by promoting the growth, angiogenesis, proliferation, migration, and formation of the CRC microenvironment through different pathways [15]. Il-6 is also a sensitive marker of inflammation, with levels increasing both locally and in plasma after tissue injury. It activates the immunosuppressive cytokines such as Il-10. The role of Il-10 in cancer pathogenesis and development is complex; Il-10 also plays an important role as an anti-inflammatory marker, inhibiting the synthesis of proinflammatory cytokines such as Il-6 [9,15]. Il-10 can repress proinflammatory responses, limiting the tissue disruptions caused by inflammation. For these reasons, we selected only patients with CRC in this study.

Cytokines have a plasmatic peek in the first 24 h after surgery; therefore, we considered studying mainly cytokines as pro- and anti-inflammatory markers, first involved in the inflammatory response of the organism. But we also analyzed the CRP and leucocytes. CRP is an indicator of the inflammatory response. Thus, it is frequently correlated with inflammatory cytokine Il-6, which may amplify CRP production. Usually, CRP increases 48 h after surgery, in response to inflammatory cytokines and is more specific for postoperative complications [16], but CRP values can change rapidly [17]. In our study we included values up to 24 h after surgery. Also, pain is more prominent immediately after surgery so the study of a correlation between pain and these markers should be implied.

Epidural analgesia remains the standard of care in the multimodal management of postoperative pain. However, this method cannot be applied to a high percentage of patients, up to 20%. In addition, when applied, the technique can be deficient or the catheter mispositioned, resulting in low efficacy. Patients with CRC are often older at the time of surgery or have associated pathologies that require anticoagulant treatment. In open colorectal surgery, the side effects of the technique include hypotension due to the sympatholytic effect with high volume filling requirements, edema of the intestinal wall, and anastomotic fistula [11].

Therefore, an alternative method of reducing postoperative pain using wound infiltration has been proposed. Different types of wound infiltration techniques have been analyzed in the literature. In this study, we combined preoperative subcutaneous infiltration, at the planned incision line, with preperitoneal infiltration just before abdominal closure and compared the pain reported with this technique with that reported for preperitoneal infiltration alone.

Pre-emptive infiltration with local anesthetic influences the development of pain through the antinociceptive effect exerted on the nerve endings at the level of the surgical incision, thus preventing peripheral and central hypersensitivity, increasing the pain threshold, and decreasing the intensity of postoperative pain [18]. Pre-emptive analgesia produces superior postoperative pain control compared with analgesic interventions performed after the development of the pain stimulus [10,18]. Other studies that analyzed only the preincisional block reported limited results regarding the reduction in postoperative pain [19]. However, as somatic pain is initiated in the deep layers of the abdominal wall, preperitoneal infiltration was found to be more efficient than the infiltration of the superficial plane [20]. Another theory suggests neural communication between the peritoneum and the brain via vagal afferents, activated by the surgical trauma and with subsequent peritoneal inflammation, which can bypass epidural analgesia [21]. Thus, infiltration with a local anesthetic in the preperitoneal space or even intraperitoneal anesthesia should have a more potentiated effect on pain, as demonstrated by Kahokehr et al. [22]. This was a reason to combine pre-emptive and preperitoneal analgesia, with the hope of enhancing pain control.

The frequency of local anesthetic administration also plays an important role in reducing pain. Continuous wound infiltration through a preperitoneal catheter seems to be superior in terms of pain control and opioid use, being therefore an alternative to epidural analgesia [8]. This technique is complex, requiring experience for implementation, and is associated with complications such as infection, leakage, kinking, and obstruction. Wound infection seems to be a particular problem addressed in studies, occurring in up to 13.8% of cases [20]. PROSPECT recommends continuous wound infiltration as a substitute for epidural analgesia in open colorectal surgery [23]. However, due to the technique’s limitations and complication risks, many researchers have opted for wound infiltration; for these reasons, we also used wound infiltration in our study as an accessible and easy-to-perform method.

In our study, the infiltration of multiple sites (subcutaneous and preperitoneal) was not necessarily followed by a more potentiated effect, as illustrated by the similar pain scores between the groups, but the method was more effective than the control method. The similar VAS values can also be explained by the rescue therapy with opioids that was used in all patients until a VAS pain score < 4 was obtained. No statistically significant difference was found regarding the opioid use between the groups.

As local anesthetic, we used ropivacaine owing to its low neurotoxicity and cardiotoxicity and long-term effects. Additionally, ropivacaine does not impact wound healing or the tensile strength of the infiltrated tissue [20]. We chose the inferior value of the normal range for ropivacaine (2–3 mg/kg body weight) to reduce the risk for systemic toxicity. The oncological patient usually presents with hypoalbuminemia and therefore the protein binding is low, with increased free fraction and higher risk of toxicity than the general population.

The surgical approach more strongly influences postoperative inflammation than the type of analgesia. Siekmann et al. found that open colorectal surgery more strongly affected cytokine (Il-6, Il-8, and Il-10) levels than laparoscopic surgery, because inflammation was not influenced by the type of analgesia [21]. In colorectal surgery, the shift toward laparoscopic techniques is associated with the substantially reduced parietal component of postoperative pain, but laparoscopic techniques remain similar to open surgery regarding the resulting acute visceral pain [20]. In laparoscopic surgery, wound infiltration alone ensures effective analgesia, demonstrated even in radical procedures, with similar results to those of other types of regional anesthesia (such as transversus abdominis plane block) in terms of pain control, amounts of analgesics used, and other short-term outcomes [24,25,26]. We selected only patients undergoing open colorectal surgery only to obtain an enhanced inflammatory response.

Local anesthetics inhibit the inflammatory response to injury, reducing the levels of locally released inflammatory mediators [20]. High levels of the proinflammatory cytokine Il-6 after surgery are expected. In our study, we registered the highest postoperative Il-6 level in the control group. As this was the only group without local infiltration or epidural analgesia, the result confirms the influence of wound infiltration in reducing the levels of inflammatory cytokines. Of the two groups receiving local infiltration, the proinflammatory cytokine levels in the preperitoneal group were closer to those in the epidural group, whereas the association with preoperative wound infiltration was not followed by lower cytokine levels.

The anti-inflammatory response to local anesthetic infiltration was strong after the surgery in patients with preperitoneal infiltration, in whom the highest Il-10 levels were recorded. A significant increase in Il-10 levels was also noticed in the control group. This result can be explained by the small number of patients in the group or by the dynamics of IL-10, which are unclear. Il-10 can have a dual role, acting as an anti- and as a proinflammatory cytokine, as supported by several studies [27,28].

The dynamics between the cytokine levels were also characterized by a proportional increase in Il-10 with Il-6 levels 6 h after surgery. The dynamics of the levels of all the markers in group 2 were similar to those in the epidural group; therefore, we concluded that the method triggered similar responses. This response was validated through the notable pain reduction, with similar pain scores 6 h after the surgery in the epidural and infiltration groups. Additionally, 6 h after surgery, we found an indirect correlation between higher Il-10 levels and lower pain scores. As an anti-inflammatory cytokine, higher values of Il-10 are consistent with a reduction in the inflammatory response and consecutively a reduction in pain. As a response to the Il-6 increase and inflammatory reaction, there is also a compensatory augmentation of the Il-10 values as an anti-inflammatory marker. This is sustained by the direct correlation between Il-6 and Il-10 values that we identified 6 h after surgery. Finaly, the more Il-10 increases, the more inflammatory response is inhibited, and the pain is reduced.

The correlation between the Il-10 levels and pain scores can be interpreted as the effect of the analgesic techniques. Wound infiltration is associated with reduced inflammatory cytokine levels and a potentiated anti-inflammatory response, which translated into reduced pain scores in these groups. However, these results did not influence the hospitalization duration or admission to the ICU in our study.

The limitations of this study are the small number of patients in each group. Because of the limited data in the literature on the subject, we performed a pilot study to evaluate possible results in this direction. Of course, larger studies are necessary to confirm these observations. Finally, a short period was analyzed (24 h after surgery), and the variations in cytokine levels can be major in the days following a surgical procedure. However, we were limited by the duration of the local anesthetic’s effect because we performed preincisional and preperitoneal infiltrations.

## 5. Conclusions

Wound infiltration techniques are associated with reduced pain scores, similar to epidural analgesia. These techniques have the potential to reduce the postoperative inflammatory response while potentiating the anti-inflammatory reaction.

Our study supports the implementation of wound infiltration techniques as part of the multimodal management of postoperative pain. Their importance is even higher in patients when other regional anesthesia techniques are contraindicated or in patients with multiple comorbidities.

Other studies are necessary to completely characterize the dynamics between wound infiltration and cytokine levels and to analyze the immuno-modulatory effects of wound infiltration in oncologic surgery.

## Figures and Tables

**Figure 1 medicina-60-01244-f001:**
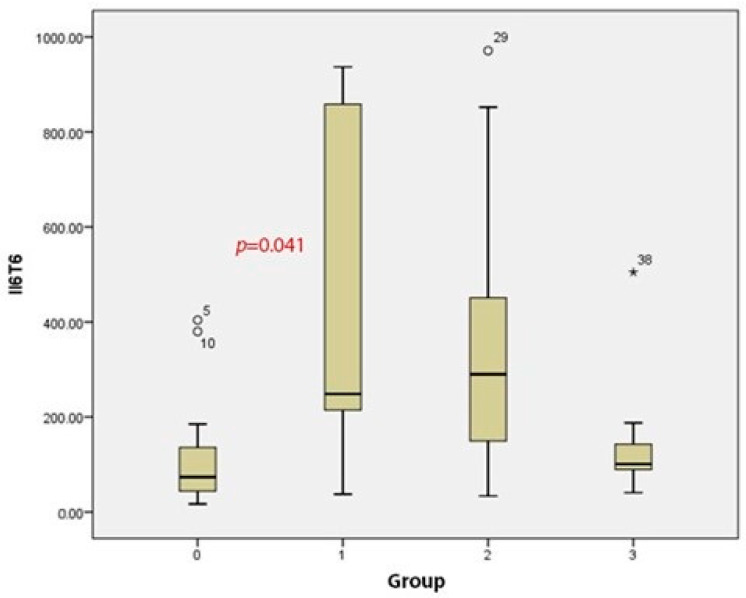
Postoperative Il-6 levels.

**Figure 2 medicina-60-01244-f002:**
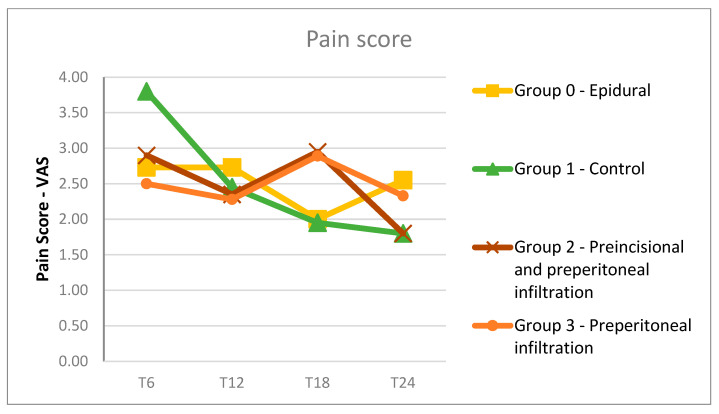
Pain scores at 6, 12, 18 and 24 h after surgery in the four groups.

**Figure 3 medicina-60-01244-f003:**
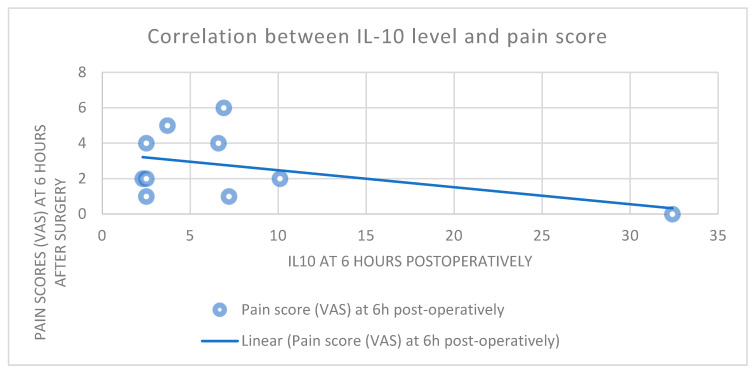
Indirect correlation between IL-10 levels and pain scores at 6 h after surgery in the preincisional and preperitoneal infiltration group.

**Figure 4 medicina-60-01244-f004:**
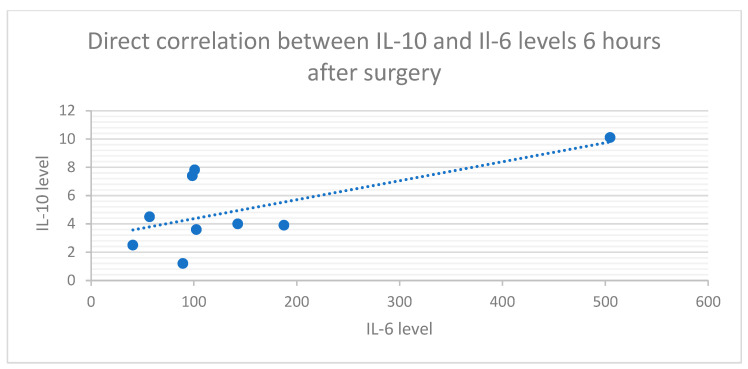
Direct correlation between IL-10 values and IL-6 values 6 h after surgery in the preperitoneal infiltration group.

**Table 1 medicina-60-01244-t001:** Demographic characteristics of the study groups.

	Group 0 (*n* = 11)	Group 1 (*n* = 10)	Group 2 (*n* = 10)	Group 3 (*n* = 9)	*p*
Age (year)	70.27 ± 7.6	72.7 ± 7.8	70.5 ± 6.2	69 ± 6.0	0.711
Sex					
Male	5 (45.4%)	6 (60%)	4 (40%)	7 (77.7%)	0.370
Female	6 (54.5%)	4 (40%)	6 (60%)	2 (22.2%)	0.370
ASA					
I	0 (0)	1 (10%)	0 (0)	0 (0)	0.404
II	3 (27.2%)	1 (10%)	2 (20%)	3 (33.3%)	0.666
III	8 (72.7%)	8 (80%)	8 (80%)	6 (66.6)	0.865
Comorbidities					
Cardiovascular disease	11 (100%)	8 (80%)	7 (70%)	5 (55.5%)	0.113
Obstructive pulmonary disease	1 (9%)	1 (10%)	1 (10%)	2 (22.2%)	0.817
Diabetes mellitus	5 (45.4%)	2 (20%)	2 (20%)	3 (33.3%)	0.549
Surgical procedure					
Right hemicolectomy	7 (63.6%)	4 (40%)	8 (80%)	4 (44.4%)	0.264
Left hemicolectomy	0 (0)	2 (20%)	1 (10%)	2 (22.2%)	0.429
Sigmoid resection	2 (18.1%)	2 (20%)	0 (0)	2 (22.2%)	0.517
Anterior rectum resection	2 (18.1%)	2 (20%)	1 (10%)	1 (11.1%)	0.910

**Table 2 medicina-60-01244-t002:** (**a**) Mean levels of Il-6, Il-10, C-reactive protein (CRP), and leukocytes in the four study groups, before (T0) and after surgery (T6). (**b**) Comparison of the levels of the biomarkers between groups and control and among all four groups, before (T0) and after surgery (T6). The bold is for statistically significant values.

(a)					
Groups		Marker	T0 (Mean ± SD)	T6 (Mean ± SD)	*p* Value
Group 1 (Control)		IL-6	9.23 ± 10.70	435.54 ± 340.36	**0.003**
		IL-10	3.38 ± 1.24	6.82 ± 4.35	**0.033**
		CRP	2.37 ± 4.36	10.62 ± 5.35	**0.000**
		Leucocite	7.45 ± 2.57	9.44 ± 2.72	0.167
Group 0		IL-6	10.09 ± 12.24	128.07 ± 138.29	**0.021**
		IL-10	3.64 ± 3.89	5.2 ± 2.75	0.78
		CRP	3.01 ± 3.33	8.26 ± 5.38	**0.012**
		Leucocite	7.68 ± 3.41	11.98 ± 3.06	**0.001**
Group 2		IL-6	49.45 ± 99.11	371.91 ± 317.24	**0.011**
		IL-10	3.69 ± 4.98	7.67 ± 9.09	0.236
		CRP	2.55 ± 3.41	8.03 ± 4.87	**0.000**
		Leucocite	6.48 ± 1.76	11.85 ± 3.76	**0.000**
Group 3		IL-6	7.41 ± 5.18	147.06 ± 140.91	**0.015**
		IL-10	2.77 ± 1.36	5.00 ± 2.84	**0.029**
		CRP	1.93 ± 1.97	10.19 ± 3.24	**0.000**
		Leucocite	8.25 ± 2.49	12.09 ± 4.64	0.078
**(b)**					
**Marker**	**Group**	**Time**	** *p* ** **Value (Group vs. Control)**	** *p* ** **Value between All Groups—T0**	** *p* ** **Value between All Groups—T6**
IL-6	0	T0	1	0.18	**0.018**
		T6	**0.041**		
	2	T0	0.44		
		T6	1		
	3	T0	1		
		T6	0.106		
IL-10	0	T0	1	0.929	0.644
		T6	1		
	2	T0	1		
		T6	1		
	3	T0	1		
		T6	1		
CRP	0	T0	1	0.917	0.534
		T6	1		
	2	T0	1		
		T6	1		
	3	T0	1		
		T6	1		
Leucocytes	0	T0	1	0.53	0.304
		T6	0.68		
	2	T0	1		
		T6	0.84		
	3	T0	1		
		T6	0.69		

**Table 3 medicina-60-01244-t003:** Lengths of ICU and hospital stays.

Variable	Group 0	Group 1	Group 2	Group 3	*p*
LHS (days)	14.64 ± 7.27	10.7 ± 3.88	10.7 ± 2.49	12.3 ± 7.26	0.33
LHS in ICU (days)	1.45 ± 1.63	1.7 ± 1.88	0.9 ± 0.99	4.5 ± 9.43	0.34

**Table 4 medicina-60-01244-t004:** Number of patients that required rescue analgesia via opioid administration in the four groups at 6, 12, 18, and 24 h after surgery.

Time after Surgery		Rescue Analgesia	No Rescue Analgesia	*p*
6 h	Group 0	2	9	0.164
	Group 1	5	5	
	Group 2	5	5	
	Group 3	6	3	
12 h	Group 0	3	8	0.098
	Group 1	3	7	
	Group 2	4	6	
	Group 3	7	2	
18 h	Group 0	3	8	0.280
	Group 1	3	7	
	Group 2	4	6	
	Group 3	6	3	
24 h	Group 0	3	8	0.165
	Group 1	2	8	
	Group 2	4	6	
	Group 3	6	3	

## Data Availability

The original contributions presented in the study are included in the article, further inquiries can be directed to the corresponding author/s.

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
