# Peer review of "The Influence of Analgesic Wound Infiltration on Postoperative Pain and Inflammatory Cytokines in Open Colorectal Surgery: A Randomized Comparative Pilot Study"

_medicina, 2024, doi:10.3390/medicina60081244_

Round 1

Reviewer 1 Report

Comments and Suggestions for Authors

I read your manuscript it's well written, however I have 2 questions

1- regarding the fact that we have IL-6 elevation but not CRP ,the importance of this data os under question ,wouldn't it be better to evaluate different inflammatory pathway markers and the common inflammatory pathway as well.

2- The fact that using other methods of anesthetic drug other than direct infiltration,didn't have adequate pain control may propose that the effective dose wasn't given. Also did the opioid or non opioid painkiller were given? 

If not is mere anesthetic infiltration for the patient justifiable, since it can't suppress peritoneal pain

Author Response

  1. CRP is an indicator of the inflammatory response. Thus, it is frequently correlated with inflammatory cytokine Il-6, which may amplify CRP production. Usually CRP increases 48h after surgery, in response to inflammatory cytokines and is more specific for postoperative complications (Selvamani Tet al. Predictors That Identify Complications Such As Anastomotic Leak in Colorectal Surgery: A Systematic Review. Cureus 2022 14(9): e28894), but CRP values can change rapidly (Wautier JL et al. Old and New Blood Markers in Human Colorectal Cancer. Int. J. Mol. Sci. 2022, 23, 12968). In our study we included values up to 24h after surgery, therefore variations after this period have not been analyzed. On the contrary, cytokines have a plasmatic peek in the first 24h after surgery so therefore we considered studying mainly cytokines as pro- and anti-inflammatory markers, first involved in the inflammatory response of the organism. Also, pain is more prominent immediately after surgery so the study of a correlation between pain and these markers should be implied.
  2. In this study we used multimodal analgesia according to hospital protocol and if pain score > 4 rescue analgesia was represented by opioids. The multimodal analgesia was represented by: Paracetamol, Dexamethasone, Ketamine, Nefopam and Metamizole. The protocol was the same in all four groups. The only variable was the method of analgesia: epidural or local anesthetic infiltration.  The dose used for Ropivacaine is 2-3 mg/kg body weight. We have chosen the inferior value of the range to reduce the risk for systemic toxicity. The oncological patient usually presents with hypoalbuminemia and therefore the protein binding is low, with increased free fraction and higher risk of toxicity than general population. We also excluded patients with organ failure (ASA risk > III) to which the administered dose should have been reduced.

Reviewer 2 Report

Comments and Suggestions for Authors

The authors present an interesting randomized clinical study, in which they analyze different preoperative anesthesia techniques and compare them with conventional epidural analgesia.

In the abstract, the following sentence should be corrected: Il-6 postoperative values were significantly lower in group 0 compared to control (p=0.041), since the comparison of the "control" group is not understood.

 The Methods section should explain why patients with ASA risk > III were excluded.  In addition, they should explain the method used to randomize these patients, as it is not clear.

The statistical analysis is adequate, and I congratulate the authors for being so clear in the statistical methods used.

Among the limitations of the study, they should insist that the small sample size of the study may bias the results. With 10 patients per group, it is difficult to draw conclusions and extrapolate the results.

Table 2 is somewhat confusing, with many different p-values, which makes it difficult to understand. I recommend modifying it to make it clearer and simplified.

In the Discussion, they should justify this statement: "at 6h after surgery, we found an indirect correlation between higher Il-10 values and lower pain scores". Inflammation leads to pain, due to the secretion of proinflammatory cytokines, so there should be a direct relationship, and not an inverse one, between cytokine levels and pain. They should elaborate on this statement, or give different plausible possibilities.

Comments on the Quality of English Language

Minor editing of English language required

Author Response

1. We changed the formulations in the abstract to make it clearer.

2. Patients with ASA risk > III were excluded because there are patients with organ failure to which the administered dose of local anesthetic should have been reduced. 

3. Patients included in the present study were randomized according to block randomization method. Block randomization has the advantage of increasing the comparability between groups by keeping the ratio of the number of subjects between groups almost the same. 

4. The data in the literature is limited to the subject and we performed this pilot study to evaluate possible results on this direction. Of course, larger studies are needed to confirm our initial results.

5. Table 2 – we have simplified the table by making 2

6. As Il-10 is an anti-inflammatory cytokine, higher values are consistent with a reduction of the inflammatory response and consecutively a reduction of pain. Il-6 is the pro-inflammatory cytokine and has significantly higher values after surgery. As response to Il-6 increase and inflammatory reaction, there is also a compensatory augmentation of the Il-10 values as an anti-inflammatory marker. This is sustained by the direct correlation between Il-6 and Il-10 values that we identified 6h after surgery. Finaly, the more Il-10 increases, the more inflammatory response is inhibited, and the pain is reduced.

We have introduced all the explanations in the article as well.